# Mapping Deprived Urban Areas Using Open Geospatial Data and Machine Learning in Africa

Maxwell Owusu [1], Ryan Engstrom [1,*], Dana Thomson [2], Monika Kuffer [2] and Michael L. Mann [1]

1. Department of Geography, The George Washington University, Washington, DC 20052, USA; mowusu@gwu.edu (M.O.); mmann1123@gwu.edu (M.L.M.)
2. Faculty of Geo-Information Science & Earth Observation (ITC), University of Twente, 7522 NH Enschede, The Netherlands; d.r.thomson@utwente.nl (D.T.); m.kuffer@utwente.nl (M.K.)
* Correspondence: rengstro@gwu.edu

**Abstract:** Reliable data on slums or deprived living conditions remain scarce in many low- and middle-income countries (LMICs). Global high-resolution maps of deprived areas are fundamental for both research- and evidence-based policies. Existing mapping methods are generally one-off studies that use proprietary commercial data or other physical or socio-economic data that are limited geographically. Open geospatial data are increasingly available for large areas; however, their unstructured nature has hindered their use in extracting useful insights to inform decision making. In this study, we demonstrate an approach to map deprived areas within and across cities using open-source geospatial data. The study tests this methodology in three African cities—Accra (Ghana), Lagos (Nigeria), and Nairobi (Kenya) using a three arc second spatial resolution. Using three machine learning classifiers, (i) models were trained and tested on individual cities to assess the scalability for large area application, (ii) city-to-city comparisons were made to assess how the models performed in new locations, and (iii) a generalized model to assess our ability to map across cities with training samples from each city was designed. Our best models achieved over 80% accuracy in all cities. The study demonstrates an inexpensive, scalable, and transferable approach to map deprived areas that outperforms existing large area methods.

**Keywords:** open geospatial; GIS; Africa; slums; deprived area; machine learning; remote sensing; OpenStreetMap; poverty mapping; informal settlements; LMIC

## 1. Introduction

Between 2014 and 2018, the global slum population increased from 23% to 24% [1]. Worse still, evidence from 2020 and 2021 indicated that slums or deprived urban population increased due to the interrelated challenges of high population growth, localized impact on climate change, COVID-19 pandemic, and economic crises [2,3]. The majority of urban residents in Africa live in deprived areas that lack basic services, and household assets, located in environmental high-risk areas, such as flood zones, which are vulnerable to climate change [4]. The World Bank estimates that nearly 574 million people will still be living below the poverty line (USD 2.15 a day) globally if no action is taken by 2030 [5].

In response to the Sustainable Development Goals (SDGs) and related agendas, the United Nations, the World Bank, non-governmental organizations, other international organizations, and local governments requires adequate information about the location and characteristics of deprived areas to "develop plans, monitor progress and consider how existing programs can better address the specific vulnerabilities of different populations" [6]. Such information includes the growth of deprived areas; socioeconomic characteristics, such as access to social services; and physical characteristics, such as the durability of building materials. This information is necessary to plan coordinated citywide renewal and development projects and to target poverty alleviation interventions [7].

Effectively planning and monitoring specifically requires timely, accurate, and city-wide data at the block or neighborhood scale. However, this information is rarely available in many low- and middle-income countries (LMICs).

Existing data are often piecemeal about single communities, inaccurate, and divorced from local context or they are outdated or aggregated at district or national level [8,9]. The four broad sources of data about deprived areas are produced by mostly siloed slum mapping traditions: (1) censuses and/or household surveys, (2) field-based mapping, (3) the manual digitization of areal/satellite imagery, and (4) the modeling of remotely sensed imagery [10]. Censuses and/or household surveys collect socioeconomic indicators that are used to characterize deprived households and are aggregated by district, city, or all urban areas nationally. Not only is this type of data expensive and resource-intensive to produce, but household-level deprivations (e.g., access to a private improved toilet) are different from area-level deprivations (e.g., whether a public sewage system exists, serves most/all households, and successfully treats all waste without leakage) [6]. Field-based mapping is primarily conducted in a participatory manner, working with individual communities to map their settlement [11]. For example, the map Kibera project in Nairobi (Kenya) [12] and more than 7000 communities across Africa, Asia, and Latin America within the Slum/Shack Dweller International network [13]. Community members collect detailed area- and household-level data to jointly define challenges and resources and use this information to plan and prioritize internal upgrading initiatives as well as to advocate for basic public services. Although community-generated data are rich with local context information, they are resource-intensive to collect, not scalable to every deprived community, and have low temporal granularity. The manual digitization of satellite imagery, for example in OpenStreetMap, by global volunteers or local experts can produce some reliable information about citywide physical characteristics (e.g., roads and building footprints), but these data are incomplete, especially in LMICs [14], and a majority of the most relevant features (e.g., water taps or "slum" boundaries themselves) are impossible to map accurately without intimate local context knowledge or field verification.

Similarly, machine learning models based on remote sensing can provide information about many of the physical characteristics of deprived areas, including building and road morphology and landcover [15,16], and they are cost-efficient compared to the methods mentioned above. However, a major limitation is that remote sensing ignores important socioeconomic indicators and requires some context knowledge and decent training data to train accurate models [17].

The "Integrated Deprived Area Mapping System (IDEAMAPS)" Network was established by members of these diverse slum mapping groups to integrate the strengths of each approach to produce routine, consistent, accurate maps of deprived areas across cities [18]. A key aspect of the project is to utilize open geospatial data and low-cost tools that are scalable and transferable in multiple cities. The use of geospatial data for deprived area mapping is yet to be fully exploited [19]. Major reasons relate to unstructured data, data quality, data completeness, and varying spatial and temporal resolutions [14,17,20]. For example, a recent study shows that the Global Human Settlement population dataset underestimates slum areas [21]. Consequently, very few studies have combined multiple socioeconomic indicators to map deprived areas. Most studies have mainly focused on the use of satellite images to map slum morphologies (see [15,22,23] for a more detailed review). One notable work from Mahabir and colleagues combined multiple geospatial socioeconomic indicators, including population density, real estate price, birth rates, access to pit latrine, and places of worship, in order to map deprived areas in Kenya [20]. Other researchers have combined remote sensing and census data [24–26]. However, these approaches are difficult to replicate in areas where spatial census data are unavailable.

The advancement of open geospatial data presents new opportunities to address deprived area mapping and account for the limitations of existing approaches [17,20]. Open geospatial data are public or private digital data with an open license that guarantees free access to data without limitation or few restrictions [27]. These sources of data are

increasingly becoming available globally due to open data initiatives by governments, NGOs, and private companies. For example, Meta and WorldPop provide high-resolution geospatial data on population and demographics at a global scale [28,29]. Other geospatial data include urban heat islands, OpenStreetMap, and Malaria Atlas [21,29,30]. Point-of-interest data from OpenStreetMap can be used for creating accessibility features, including proximity to infrastructure, such as health, schools, and employment opportunities [31]. The advantage of open geospatial datasets is that they guarantee free access, allowing for the reuse and reproducibility of methods. They offer an alternative or complementary data source to map and characterize deprived areas both physically and socially. They can potentially provide new insight into deprived area living conditions by adding equally important socioeconomic features often missing in remote sensing-based methods [15,17,22].

In this study, we harmonized and extracted physical and socioeconomic indicators from open geospatial data to map deprived areas. We developed and tested a machine learning model to map and characterize deprived areas in multiple cities and at a large scale. The study operationalized the recently published IDEAMAPS domain of deprivation frameworks to (i) identify relevant indicators and corresponding geospatial layers for characterizing deprived areas, (ii) process the identified geospatial layers in a systematic and consistent manner for machine learning modeling, and (iii) analyze the relative importance of the indicators for modeling. The IDEAMAPS domain of deprivation framework provides a holistic perspective on social and physical characteristics that help to define urban deprivation across a variety of contexts, including indicators from traditional sources (e.g., surveys) and big data (e.g., social media). Yet, the framework has not been applied on a large scale. The main contribution of this study includes:

- Landscape analysis to identify relevant indicators with corresponding geospatial data that are globally available for modeling.
- Designed and tested machine learning models to predict and characterize deprived areas in multiple cities and on a large scale.
- Analyzed the relative importance of indicators for global mapping. This allows us to know the most relevant indicators as we aim for global mapping.

## 2. Study Area

The study was conducted in three Sub-Saharan African cities—Accra (Ghana), Lagos (Nigeria), and Nairobi (Kenya) as shown in Figure 1. These were the cities that the IDEAMAPS project focused on, and we have existing local networks providing us with reference data and local context knowledge. The three cities also allow us to test across a variety of urban morphological characteristics and social and environmental contexts. Moreover, they have a large segment of the population in deprived living conditions [32–34]. City boundaries or region of interest (ROI)—a spatial context that is of interest for the study—were defined using the intersection of their administrative boundaries and the larger built-up area as defined with WorldPop 100 × 100 m building footprint metrics [35,36], with an added 1 km buffer to ensure the inclusion of peri-urban deprived areas and to test the scalability of our method.

Accra is a coastal city and the economic hub of Ghana, with a population of approximately four million [37]. Unplanned urbanization has led to a housing backlog, with 34% of residents in the inner city (less than 5% of land area) living in slums [34]. Accra has expanded to include suburbs such as Ashiaman, Tema, Kosoa, and Nasawan. We chose an ROI of 2561 square kilometers to cover both the main city and the peri-urban area. Lagos is a port city with wetlands covering 22% of the land area. In 2021, it was estimated that over 20 million residents in Lagos live in slums [38]. We chose an ROI of 5638 square kilometers, which is larger than the main Lagos state administrative areas covering 3345 square kilometers. Nairobi, the capital of Kenya, has approximately 60% of residents living in slums [39]. The chosen ROI of 2006 square kilometers, which goes beyond the administrative area, covering 695 square kilometers.

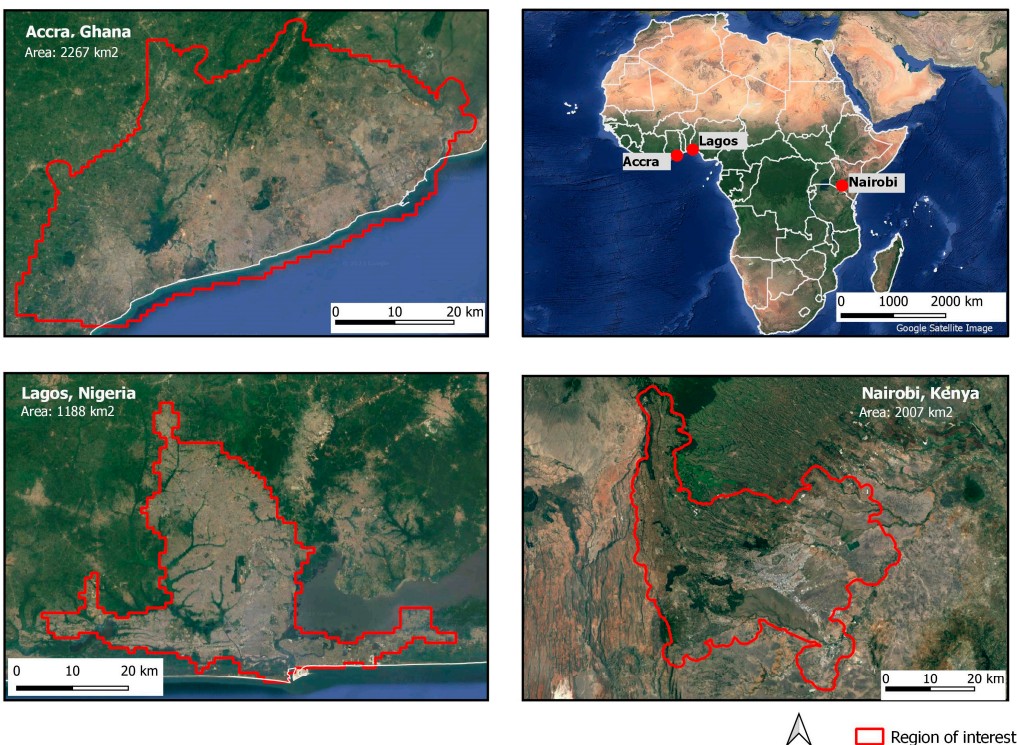

**Figure 1.** Map shows the citywide boundary used for the study. The study area consists of three Sub-Saharan cities—(1) Accra, Ghana; (2) Lagos, Nigeria; and (3) Nairobi, Kenya. Imagery source: Google Satellite Image.

## 3. Materials and Methods

To build a scalable and transferable approach to map and characterize deprived areas in multiple cities, a four-phase step with each phase building on the findings from the previous was used (Figure 2). Phase 1 conceptualized an operational definition and relevant indicators for area-based mapping that integrates the physical and social characteristics of deprived areas based on the IDEAMAPS deprivation framework [40]. Phase 2 processed identified geospatial layers in a consistent and systematic manner to integrate different data sources with varying temporal and spatial resolution. This provides us with a dataset ready for modeling. Phase 3 designed and tested a scalable and transferable machine learning model to predict deprived areas and assessed the accuracy of the model. Phase 4 analyzed the relative importance of the indicators used for modeling.

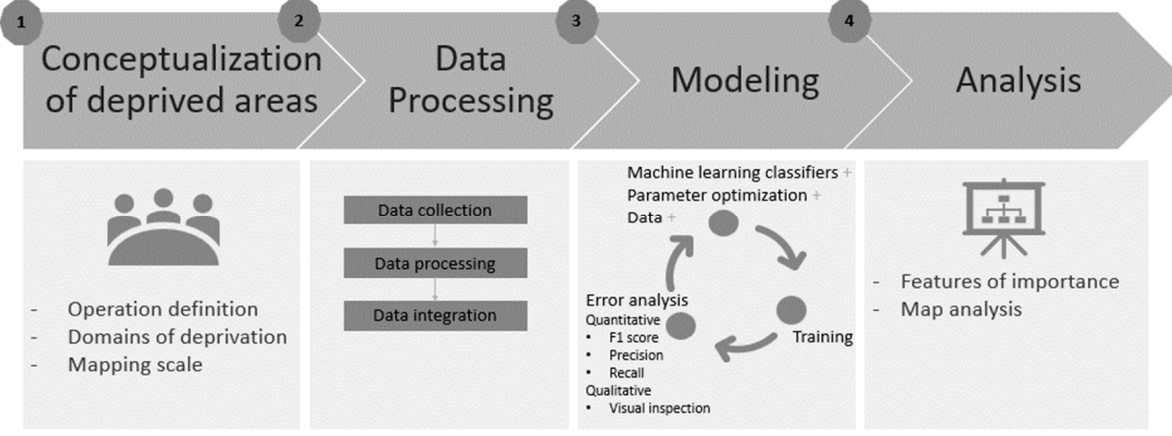

**Figure 2.** Four-phase steps of the methodology of the study.

Across all the phases, the team coordinated with the IDEAMAPS network, and experts engaged in urban deprivation-related works to share findings to highlight commonality and amplify the finding of the study. The experts were selected through purposive sampling and comprised a diverse group, including researchers from institutions such as Center for Urban and Environmental Research (CUER) of George Washington University, the Faculty of Geo-information Science and Earth Observation at the University of Twente, the Urban Big Data Center of the University of Glasgow, the University of Lagos, and the African Population and Health Research Center as well as experts from NGOs such as the Justice & Empowerment Initiatives in Nigeria and the People's Dialogue in Ghana.

### 3.1. Conceptualizing Deprived Areas

The challenge of the global mapping of deprived areas starts with conceptualizing an operational definition. The often-used terminology is a slum or "informal settlements". These terms are conceptually ambiguous. They have different definitions and are known by different local names for several reasons, such as cultural context and available building materials and infrastructure [41], as can be seen, for example, in Zongo in Ghana, Favelas in Rio De Janeiro, Kachi Abadi in Karachi, and Vijiji in Nairobi. UN Habitat defines slums at the household level. A slum is any household that lacks one of the following: access to water, improved sanitation, durable housing, sufficient living space, and tenure security [42]. The slum household definition ignores area-based risks such as crime, flood risks, and lack of social amenities that residents face that occur outside of the household. Other associated problems are that slums are conceptually relative as every country has its own definition of slums [10], and it is widely criticized as it can denote bad connotations for stigmatization and forced eviction [18,43].

Due to the conceptual complexities of slums, the study draws inspiration from existing studies on measuring deprivation or "deprived area" at the area-based level, particularly Thomson and colleagues [18]. The term "deprived area" is widely used by Earth observation (EO) scientists as it focuses on area-level deprivation [10]. EO scientists use morphological characteristics, including building size, density, and settlement pattern, to distinguish deprived areas from non-deprived areas [44]. For this study, deprived areas are defined as urban spaces that lack physical and social assets, is often a result of unplanned urbanization, is prone to disaster, and is characterized by poverty and substandard living conditions. This operation definition integrates the physical and social characteristics of deprived areas. It provides a broad view of deprived areas and new insight into their characteristics.

Drawing from the IDEAMAPS deprivation framework [40], we identified six domains—(1) physical hazards, (2) unplanned urbanization, (3) population characteristics and housing, (4) social hazards, and (5) facilities and services. Physical hazard relates to the exposure to risk. Indicators include flood zones, steep slopes, pollution, lack of vegetation, heat stress, proximity to roads, wetlands, rivers, railways, hazards industry, and high-voltage power lines. Unplanned urbanization is associated with slum-like characteristics such as irregular settlement shape and patterns, high building density, lack of roads, small building size, and poor building materials. Population characteristics and housing relate to demographic and health characteristics. Indicators include high population count/density, poor housing conditions, and ethnicity. Social hazards relate to the social risk of a neighborhood. Indicators include high crimes, unsafe neighborhoods, unmet needs for family planning, risk of disease outbreaks, and ethnolinguistic groups. Facilities and services relate to a dweller's access to infrastructure and social services. Indicators include access to health, water and sanitation, electricity, financial services, education, and recreational facilities.

### 3.2. Geospatial Indicators

Based on suggestions by experts, we drew on recent open geospatial data that are available for the three cities over a timeframe spanning from 2010 to 2022. Using a snowballing approach, we expanded the list of data. In addition, we conducted a targeted search on

organizations known to be interested in deprivation. Such organizations include WorldPop, CIESIN, and NASA. We tried to find the most recent data with complete coverage for the entire study area. Table 1 presents the indicator, data source, year, and description.

**Table 1.** Geospatial indicators and their description. In blue are the user-defined features in Section 3.4 below.

| | Domains | Indicator | Data Source | Year | Description | Original Spatial Resolution |
|---|---|---|---|---|---|---|
| 1 | Facilities and service | Distance to health facility | Population Health Unit, Kenya Medical Research Institute—Wellcome Trust Research Programme | 2019 | Distance to health facility | - |
| 2 | Facilities and service | Distance to major road | OpenStreetMap and WorldPop | 2016 | Distance to major road | 100 m |
| 3 | Facilities and service | Distance to road intersection | OpenStreetMap and WorldPop | 2016 | Distance to major road intersections | 100 m |
| 4 | Facilities and service | Distance to major waterway | OpenStreetMap and WorldPop | 2016 | Distance to major waterways | 100 m |
| 5 | Facilities and service | Distance to minority religious facility | OpenStreetMap | 2019 | Distance to minority religious facility (compared to city average). | - |
| 6 | Facilities and service | Distance to religious facilities | OpenStreetMap | 2019 | Distance to religious facilities | - |
| 7 | Facilities and service | Distance to government office | OpenStreetMap | 2022 | Distance to government office | - |
| 8 | Facilities and service | Access to Finance | HDX and OpenStreetMap | 2020 | Distance to finance | - |
| 9 | Facilities and service | Access to School | HDX and OpenStreetMap | 2020 | Distance to education facility | - |
| 10 | Housing | Improve housing prevalence | The Malaria Atlas Project | 2015 | Improved housing prevalence | 5 km |
| 11 | Physical hazard | Distance to river | WWF HydroSHEDS | 2007 | Distance to river | 15 arc-second resolution |
| 12 | Physical hazard | Night light | WorldPop | 2012–2016 | VIIRS night-time lights between 2012 and 2016 | 100 m |
| 13 | Physical hazard | Distance to aquatic vegetation | WorldPop | 2015 | Distance to ESA-CCI-LC aquatic vegetation area edges | 100 m |
| 14 | Physical hazard | Distance to artificial surface | WorldPop | 2015 | Distance to ESA-CCI-LC artificial surface edges | 100 m |
| 15 | Physical hazard | Distance to bare area | WorldPop | 2015 | Distance to ESA-CCI-LC bare area edge | 100 m |
| 16 | Physical hazard | Distance to cultivated area | WorldPop | 2015 | Distance to ESA-CCI-LC cultivated area edges 2015 | 100 m |
| 17 | Physical hazard | Distance to herbaceous area | WorldPop | 2015 | Distance to ESA-CCI-LC herbaceous area edges 2015 | 100 m |
| 18 | Physical hazard | Distance to inland water | WorldPop | 2018 | Distance to ESA-CCI-LC inland water (2000–2018) | 100 m |
| 19 | Physical hazard | Distance to open water coastline | WorldPop | 2020 | Distance to open-water coastline | 100 m |
| 20 | Physical hazard | Distance to shrub area | WorldPop | 2015 | Distance to ESA-CCI-LC shrub area edges 2015 | 100 m |
| 21 | Physical hazard | Distance to sparse vegetation | WorldPop | 2015 | Distance to ESA-CCI-LC sparse vegetation area edges 2015 | 100 m |
| 22 | Physical hazard | Distance to woody tree area | WorldPop | 2015 | Distance to ESA-CCI-LC woody-tree area edges 2015 | 100 m |
| 23 | Physical hazard | Slope | WorldPop | 2018 | STRM -based slope | 100 m |
| 24 | Physical hazard | Water stress | World Resource Institute (WRI) | 2010 | Baseline water stress score | 5 × 5 arc minute grid cells |
| 25 | Physical hazard | Ground water stress | World Resource Institute (WRI) | 2012 | Ground water stress score | 5 × 5 arc minute grid cells |

**Table 1.** *Cont.*

| | Domains | Indicator | Data Source | Year | Description | Original Spatial Resolution |
|---|---|---|---|---|---|---|
| 26 | Physical hazard | Hazard index | UNEP/DEWA/GRID-Europe | 2011 | This dataset includes an estimate of the global risk induced by multiple hazards (tropical cyclone, flood and landslide induced by precipitations). Unit is estimated risk index from 1 (low) to 5 (extreme). It was modeled using global data. | 1 km |
| 27 | Physical hazard | Air pollution | NASA Socioeconomic Data and Applications Center (SEDAC) | 2016 | The annual concentrations (micrograms per cubic meter) of ground-level fine particulate matter (PM2.5) with dust and sea-salt removed in 2016 | 50 m |
| 28 | Physical hazard | Biodiversity | GLOBIO | 2015 | Biodiversity (mean species abundance) | 10 arc-second |
| 29 | Physical hazard | Land cover 2 | GlobeLand30 | 2019 | GlobeLand30 includes 10 land cover classes in total, namely cultivated land, forest, grassland, shrubland, wetland, water bodies, tundra, artificial surface, bare land, perennial snow and ice. | 30 m |
| 30 | Physical hazard | Normalized Difference Vegetation Index | Desert Research Center, University of Idaho | 2019 | Maximum Normalized Difference Vegetation Index | 30 m |
| 31 | Physical hazard | Land cover 1 | Copernicus Global Land Service | 2019 | Annual 100 m global land cover maps of 2015 to 2019, generated by Copernicus Global Land service | 100 m |
| 32 | Physical hazard | Maximum ground temperature | Climatology Lab | 2019 | Maximum ground temperature | 4 km |
| 33 | Physical hazard | Multihazard distribution | CIESIN | 2005 | The Global Multihazard Frequency and Distribution is a 2.5 min grid presenting a simple multihazard index based solely on summated single-hazard decile values. | 2.5 min grid |
| 34 | Physical hazard | Climate risk | CHIRPS | 2020 | Average annual climate risk. Rainfall Estimates from Rain Gauge and Satellite Observations | 0.05 × 0.05 degree |
| 35 | Population | Population count | WorldPop | 2020 | Estimated Population Count 2020 in 100 m grid (WorldPop-UNadj-constrained) | 100 m |
| 36 | Population | Population count | Meta & CIESIN | 2018 | Estimated Population Count 2018 (HRSL-Facebook) | 1 arc-second |
| 37 | Social hazard | Pregnancy rate | WorldPop | 2017 | Estimated distributions of pregnancies | 100 m |
| 38 | Social hazard | Children with Plasmodium falciparum parasite rate | The Malaria Atlas Project | 2017 | Mean Plasmodium falciparum parasite rate in 2–10 year olds. Children with Plasmodium falciparum parasite rate | 5 km |
| 39 | Social hazard | Pregnant women antenatal care visit | Spatial Data Repository | 2014- 2016 | DHS modeled surface 2014. Percentage of women who had a live birth in the five (or three) years preceding the survey who had 4+ antenatal care visits. | 5 km |
| 40 | Social hazard | Child stunted | Spatial Data Repository | 2014–2016 | DHS modeled surface 2014. Percentage of children stunted (below −2 SD of height for age according to the WHO standard). | 5 km |
| 41 | Social hazard | DPT3 vaccine | Spatial Data Repository | 2014–2016 | DHS modeled surface 2018. Percentage of children 12–23 months who had received DPT3 vaccination. | 5 km |
| 42 | Social hazard | Delivery at health Facility | Spatial Data Repository | 2014–2016 | DHS modeled surface 2014. Percentage of live births in the five (or three) years preceding the survey delivered at a health facility. | 5 km |

**Table 1.** *Cont.*

|  | Domains | Indicator | Data Source | Year | Description | Original Spatial Resolution |
|---|---|---|---|---|---|---|
| 43 | Social hazard | Household with improve water source | Spatial Data Repository | 2014 | DHS modeled surface 2018. Percentage of the de jure population living in households whose main source of drinking water is an improved source. | 5 km |
| 44 | Social hazard | Household with insecticide-treated bednet | Spatial Data Repository | 2014 | DHS modeled surface 2018. Percentage of the de facto household population who could sleep under an ITN if each ITN in the household were used by up to two people. | 5 km |
| 45 | Social hazard | Men literature rate | Spatial Data Repository | 2014 | DHS modeled surface 2018. Percentage of men who are literate. | 5 km |
| 46 | Social hazard | Children receiving measles vaccine | Spatial Data Repository | 2014 | DHS modeled surface 2014. Percentage of children 12–23 months who had received Measles vaccination. | 5 km |
| 47 | Social hazard | Household using open defecation | Spatial Data Repository | 2014 | DHS modeled surface 2014. Percentage of the de jure population living in households whose main type of toilet facility is no facility (open defecation). | 5 km |
| 48 | Social hazard | Unmet need for family planning | Spatial Data Repository | 2014 | DHS modeled surface 2014. Percentage of currently married or in union women with an unmet need for family planning. | 5 km |
| 49 | Social hazard | Women literacy rate | Spatial Data Repository | 2016 | DHS modeled surface 2014. Percentage of women who are literate. | 5 km |
| 50 | Social hazard | Ethno-linguistic group | IMB | 2020 | Number of ethno-linguistic groups in 100 m cell. | 100 m |
| 51 | unplanned urbanization | Building count | WorldPop | 2020 | Counts of buildings that fall within 100 m grid cell. | 100 m |
| 52 | unplanned urbanization | Building density | WorldPop | 2020 | Measure of the number of buildings per grid cell area. | 100 m |
| 53 | unplanned urbanization | Rural/urban classification | WorldPop | 2018 | Urban/rural classification based on building patterns in that area. | 100 m |

### 3.3. Geospatial Layer Production

Geospatial layers were processed and resampled to a 3 arc-second (0.00083333333 decimal degree or approximately 100 m) spatial resolution to harmonize data for machine learning modeling. The 3 arc-second pixel grid offers a reasonable spatial resolution that allows the rationalization and harmonization of datasets with varying resolutions useful for modeling and analysis. It allows a balance between spatial accuracy and data integration from different sources [29]. It also offers a reasonable storage and computational overhead for large area and global scale analysis [45]. In addition, it is crucial to acknowledge that mapping deprived areas is a sociotechnical problem fraught with the risk of unintended consequences. Therefore, the deliberate choice of 3 arc-second adheres to ethical considerations by minimizing the identification of areas at-risk for eviction.

The following criteria were used to select geospatial data to be included.

1.   Must cover the region of interest.
2.   Must be either vector or raster spatial data.
3.   Must be as fine a spatial resolution as possible, usually 100 m or finer.
4.   Temporal resolution must be as close as possible.
5.   Must be available for all three cities.

Figure 3 shows the workflow for standardizing and integrating geospatial layers. It involves standardizing and resampling both raster and vector geospatial data to achieve a uniform 100 m spatial resolution. For raster data, we standardized and resampled datasets to a grid at 100 m spatial resolution.

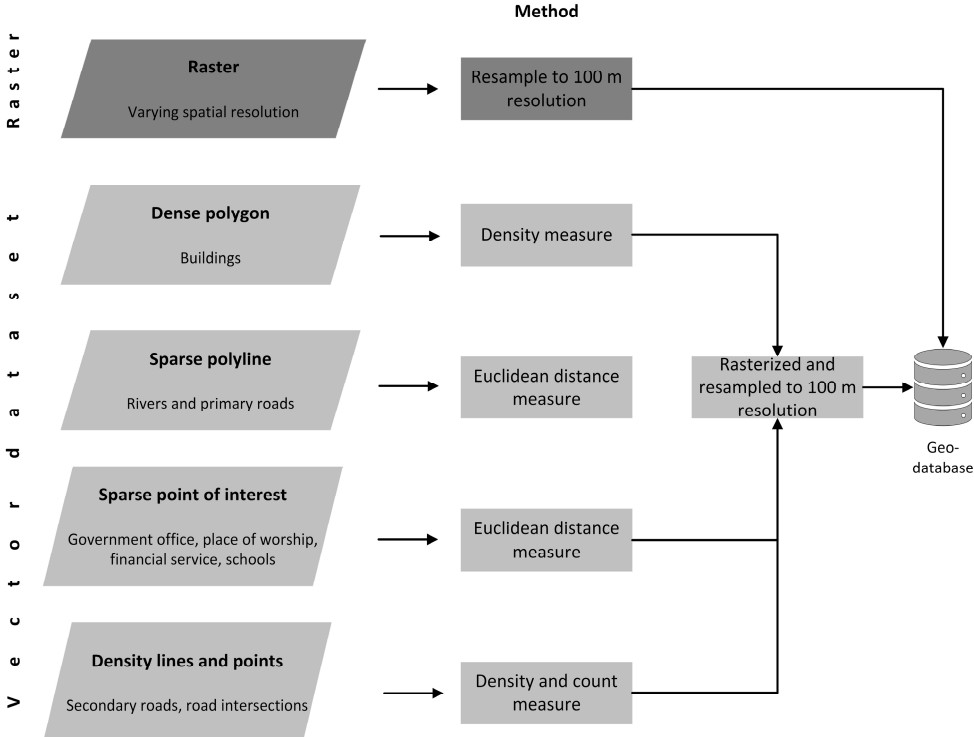

**Figure 3.** Schematic overview of workflow to produce standardized geospatial layers for modeling.

For vector data (point, polyline, polygon), we used aggregation (e.g., count or density) and accessibility methods (e.g., Euclidean distance measurement). The study used a similar workflow as discussed in [45,46]. Dense polygons, such as buildings, used the density measurement; sparse polylines, such as rivers and primary roads, used Euclidean distance measures; sparse points of interest, such as government offices, places of worship, schools, and banks, used Euclidean distance measures; and dense points or polyline, such as road intersection nodes and secondary and tertiary roads, used density and count measures. The vector outputs were rasterized and resampled to the same grid at 100 m spatial resolution. All data were reprojected to the same coordinate reference system (Mollweide projection) to ensure accurate overlay.

### 3.4. Input Features for Modeling

We experimented with two sets of input features. The first input feature set used all 53 features. This was to test models' ability to make predictions with large input data sets with varying ranges of quality. It was also to identify new insights about deprived areas and to determine whether all of these data were needed.

The second set was a manual selection of 11 features after a detailed exploratory analysis by experts and a visual assessment of the data quality for the ROIs. These are called user-defined features in the rest of the analysis. The user-defined features include building count, building density, climate risk (rainfall), improved housing, maximum ground temperature, population count, slope, urban/rural classification, and night-time light. The choice of manual selection of features allowed slum experts to leverage their expertise and understanding of the problem. Manual selection helps to select features that are informative and high quality and lead to more interpretable models. We acknowledge that ML feature selection approaches have received much attention currently [47]. However, we also observed that training samples deriving mainly from the inner city could potentially introduce some biases in the feature selection process, given our goal of mapping a large area. Also, a large portions of our ROI training and validation data were derived from outside the inner cities.

### 3.5. Classification Scheme and Sampling

The study used a binary classification scheme—deprived and non-deprived. The deprived class includes "slums", informal settlements, and areas with deprived living conditions. Reference data for the deprived class were obtained from several sources with different operation definitions and temporal resolution. These sources include Slum Dwellers International, Field Data from IDEAMAPS, Frontier Development Lab, and Mahabir GitHub repository. Therefore, the visual assessment by slum mapping experts using Google Satellite Image and Google Street View Image, together with local expert knowledge, were used to manually create additional deprived class reference data. The digitization was implemented in Google Earth Pro—a free software that allows the visualization, assessment, creation and overlay of GIS data [48]. Only areas of agreement between slum experts and local experts were used for modeling.

The non-deprived class includes non-deprived built-up areas such as formal residential, commercial, industrial, recreational. The non-built-up class includes vegetation, open spaces, waterbodies, undeveloped lands, forest, and agricultural lands. Reference data were mainly obtained from OpenStreetMap (OSM). Layers from OSM were overlaid on Google Satellite images and visually assessed to ensure that only accuracy samples were used. All sampled data were obtained in 2022.

We used tiles of 600 × 600 m for Lagos, 800 × 800 m for Nairobi and 2000 × 2000 m for Accra to create training samples. These tiles were purposely taken from different areas of the city, capturing varying deprived and non-deprived types in relative proportions. The tile sizes vary due to the different sizes of deprived areas across cities and to minimize class imbalance and the impact of spatial autocorrelation [49]. The samples were rasterized to a grid of 100 m spatial resolution. Table 2 shows the 100 m pixel counts of reference data for each city.

**Table 2.** Pixel counts of reference data for each city.

| Class | Accra | Lagos | Nairobi |
| --- | --- | --- | --- |
| Deprived | 1740 | 480 | 1300 |
| Non-deprived | 5080 | 600 | 2650 |
| Total | 6820 | 1080 | 3950 |

Random sampling was used to split tiles into training, validation, and testing to ensure unbiased estimation [50]. We randomly split the tiles into 60% for training and 40% for testing to test the generalizability of the model. We further split training tiles into 70% for training and 30% for validation. The validation tiles were used for tuning hyper-parameters to optimize models. Meanwhile, the test dataset was never seen by the models to ensure that the statistical results presented later were true to the real world.

For the generalized model, stratified random sampling was used to split tiles into training, validation, and testing. Stratification was carried out at the city level with 60% for training and 40% for testing. We further split training tiles to 70% for training and 30% for validation.

### 3.6. Modeling

Three machine learning algorithms were used for the classification task—random forest (RF) [51], multi-layer perceptron (MLP) [52] and extreme gradient boosting (XGBoost) [53]. These classification methods have achieved high predictive accuracy in land use mapping and slum mapping [54,55]. RF requires the definition of number of trees (ntree) and number of input features (mtry) to be considered at each node split. Random forest is relatively user-friendly and tends to achieve high accuracy in classification tasks [56]. It can handle large data dimensionality and deal with overfitting. MLP is a feedforward neural network that transmits from input layer to output layer in a forward direction [57]. It is relatively simple to use with fewer parameters and can work on large data sets. XGBoost

is a greedy function approximation, thus minimizing errors and able to capture complex patterns in data [58]. It has been proven to provide state-of-the-art results on many classification tasks and is suitable for handling sparse data [59]. Traditional boosting algorithms are computationally slow and often not suitable for large-area mapping. XGBoost has bridged the gap. They are efficient and flexible for processing large datasets [53]. They are relatively user-friendly, require few parameters, are easy to obtain features of importance, and have high prediction accuracy. Moreover, they are open-source and computationally fast, making them suitable for large-area mapping.

Models were trained and tested in each individual city. This helps to assess the scalability of models for large-area applications. Second, we tested a city-to-city model to assess how models perform in new geographical, morphological, social, and environmental contexts. Lastly, we developed a generalized model where training samples from each of the three cities were combined and used for training and evaluation. This allows us to assess our ability to map across individual cities, assuming we have training samples from each city.

### 3.7. Model Evaluation

To assess the classification performance, both qualitative and quantitative measures were employed. The quality measure includes a thorough visual inspection of the classification by the research team. The classified maps were overlaid on high-resolution satellite images from Google and visually inspected. We visually assessed the model's ability to map the identified deprived types in different areas where we have limited reference data.

To ensure consistency in the visual assessment, we collaborated with slum experts to define five deprived types. These deprived types integrate ground-level information from Google Street View and high-resolution aerial imagery from Google Earth Pro [48]. These deprived areas are as follows: unstructured large, structured large, unstructured small, unstructured mix, and pocket slums. Briefly, unstructured large areas have irregular patterns with relatively large roofs, mostly iron sheets, usually containing multifamily house types and concrete building material (e.g., for walls). Structured large areas have regular patterns with large roofs, mainly iron sheets, multifamily house types and concrete materials. Unstructured small areas have irregular patterns with small roofs, mainly iron sheets, detached houses, iron sheets and wooden walling materials. Unstructured mix areas have irregular patterns with a mix of large and small roofs, mix of multi-family, detached, and semi-detached house types, a mix of concrete, iron sheets, and wooden walling materials. Pocket slums have irregular patterns with very small building types (usually kiosks), temporal in nature, which comprise iron sheet and wood building materials. Details on the deprivation types can be found in [60].

The quantitative measure includes precision, recall, deprived F1-score, and macro-F1-score. Recall measures how efficiently the model retrieves a class defined as deprived [61]. Precision measures the reliability of deprived areas that are detected [62]. The macro-F1-score calculates the unweighted mean of the F1 scores calculated per class, while the F1-score represents the harmonic mean between the precision and recall of the deprived class [63]. These metrics were used to minimize the class imbalance effect and the accurate assessment of the independent test set. A default 50:50 probability threshold was used for the quantitative assessment.

### 3.8. Analysis of Importance Features

The study investigated which features are important and how much they contribute to the model. This was carried out to determine which features to continue collecting in order to support global deprived area mapping. The study used Gini impurity for computing important features and analyzed their relative contributions. Gini impurity measures the degree of how often a randomly chosen sample from the target class would be incorrectly classified if it were randomly labeled according to the distribution of the labels in the

subset [51]. It has been used in several machine learning urban classification tasks [47]. It provides good precise estimations with low model assessments.

## 4. Results

In the results section, we first present the quantitative error metrics on the testing set, then a visual assessment of best performing models is described, and lastly, the important features and how they affect the model are analyzed.

### 4.1. Quantitative Model Performance

#### 4.1.1. Individual City

The study first trains a model for each city to reflects the unique urban fabric. Table 3 provides the precision, recall, F1-deprived, and F1-macro scores for each city, model, and input feature set on the test dataset. RF achieved the best classification score for all the cities except for Lagos in all features model. F1-deprived scores are generally high in Accra and Nairobi with over 72%. Lagos achieved the lowest F1-deprived score of 65%. The poor performance of Lagos may be due to the widespread socioeconomic deprivation throughout the entire city. Most of the city of Lagos is built on wetlands and lacks services and infrastructure [38]. It is worth noting that the overall performance of using all features and user-defined features have little difference in terms of statistical accuracy. This implies that using a small input feature dataset can also achieve a good performance compared to using all features.

**Table 3.** Metrics for individual models.

| City | Input Feature Set | Model | Precision | Recall | F1-Deprived | F1-Macro |
|------|-------------------|-------|-----------|--------|-------------|----------|
| Accra | All | RF | 0.68 | 0.88 | 0.77 | 0.74 |
| | | MLP | 0.64 | 0.92 | 0.75 | 0.70 |
| | | XGBoost | 0.79 | 0.47 | 0.59 | 0.73 |
| | User-defined | RF | 0.71 | 0.84 | 0.77 | 0.78 |
| | | MLP | 0.67 | 0.76 | 0.71 | 0.78 |
| | | XGBoost | 0.78 | 0.59 | 0.67 | 0.62 |
| Lagos | All | RF | 0.80 | 0.51 | 0.62 | 0.72 |
| | | MLP | 0.84 | 0.53 | 0.65 | 0.72 |
| | | XGBoost | 0.81 | 0.33 | 0.47 | 0.62 |
| | User-defined | RF | 0.45 | 0.80 | 0.86 | 0.71 |
| | | MLP | 0.57 | 0.80 | 0.67 | 0.72 |
| | | XGBoost | 0.98 | 0.53 | 0.70 | 0.78 |
| Nairobi | All | RF | 0.81 | 0.64 | 0.72 | 0.78 |
| | | MLP | 0.56 | 0.36 | 0.44 | 0.73 |
| | | XGBoost | 0.79 | 0.46 | 0.58 | 0.68 |
| | User-defined | RF | 0.78 | 0.76 | 0.77 | 0.78 |
| | | MLP | 0.74 | 0.73 | 0.73 | 0.73 |
| | | XGBoost | 0.77 | 0.73 | 0.74 | 0.74 |

#### 4.1.2. City to City Model

In this step, we examined how a model trained in one city can detect deprived areas in other cities. This was carried out by taking the models derived for each city as described previously and then examining the results of these models' using data for the other cities. This allows us to determine the spatial transferability of models to other cities. Table 4 provides the results of the input feature set and each classifier. Except for Nairobi to Lagos, models trained in one city were able to predict deprived areas in another city with an F1-deprived score of over 60%. We observed that user-defined features achieved the highest FI deprived in all the cities. This indicates that using a few good input feature datasets is crucial to the performance of machine learning in terms of accuracy. Moreover,

it may be confusing to models when we use too many features with limited training and validation data to fine-tune them. The model trained in Lagos was able to predict deprived areas in Nairobi but not the reverse. The model trained in Nairobi achieved the lowest F1 deprived score (57%) when tested on Lagos. This indicates that models are sensitive to the physical and socioeconomic features of each city and require a detailed understating of these features to improve model results.

**Table 4.** City-to-city model.

| City | Test City | Input Feature Set | Model | Precision | Recall | F1-Deprived | F1-Macro |
|------|-----------|-------------------|-------|-----------|--------|-------------|----------|
| Accra | Lagos | All | RF | 0.89 | 0.37 | 0.52 | 0.66 |
| | | | MLP | 0.39 | 0.40 | 0.40 | 0.47 |
| | | | XGBoost | 0.82 | 0.38 | 0.52 | 0.65 |
| | Lagos | User-defined | RF | 0.78 | 0.56 | 0.65 | 0.73 |
| | | | MLP | 0.95 | 0.01 | 0.02 | 0.38 |
| | | | XGBoost | 0.81 | 0.39 | 0.52 | 0.70 |
| | Nairobi | All | RF | 0.80 | 0.48 | 0.60 | 0.76 |
| | | | MLP | 0.81 | 0.01 | 0.02 | 0.40 |
| | | | XGBoost | 0.69 | 0.58 | 0.63 | 0.72 |
| | Nairobi | User-defined | RF | 0.78 | 0.54 | 0.64 | 0.73 |
| | | | MLP | 0.31 | 0.02 | 0.04 | 0.38 |
| | | | XGBoost | 0.84 | 0.24 | 0.37 | 0.59 |
| Lagos | Accra | All | RF | 0.52 | 0.53 | 0.52 | 0.69 |
| | | | MLP | 0.40 | 0.64 | 0.50 | 0.64 |
| | | | XGBoost | 0.46 | 0.87 | 0.61 | 0.70 |
| | Accra | User-defined | RF | 0.50 | 0.92 | 0.64 | 0.73 |
| | | | MLP | 0.43 | 0.55 | 0.48 | 0.65 |
| | | | XGBoost | 0.48 | 0.85 | 0.62 | 0.71 |
| | Nairobi | All | RF | 0.78 | 0.48 | 0.59 | 0.71 |
| | | | MLP | 0.39 | 0.62 | 0.48 | 0.62 |
| | | | XGBoost | 0.73 | 0.63 | 0.68 | 0.75 |
| | Nairobi | User-defined | RF | 0.68 | 0.70 | 0.69 | 0.75 |
| | | | MLP | 0.95 | 0.02 | 0.04 | 0.40 |
| | | | XGBoost | 0.66 | 0.70 | 0.68 | 0.74 |
| Nairobi | Accra | All | RF | 0.64 | 0.61 | 0.63 | 0.76 |
| | | | MLP | 0.31 | 0.34 | 0.32 | 0.55 |
| | | | XGBoost | 0.27 | 0.49 | 0.35 | 0.51 |
| | Accra | User-defined | RF | 0.44 | 0.87 | 0.59 | 0.68 |
| | | | MLP | 0.30 | 0.61 | 0.40 | 0.53 |
| | | | XGBoost | 0.36 | 0.17 | 0.23 | 0.53 |
| | Lagos | All | RF | 0.73 | 0.07 | 0.13 | 0.43 |
| | | | MLP | 0.34 | 0.02 | 0.09 | 0.37 |
| | | | XGBoost | 0.50 | 0.24 | 0.33 | 0.51 |
| | Lagos | User-defined | RF | 0.60 | 0.25 | 0.35 | 0.54 |
| | | | MLP | 0.43 | 0.87 | 0.57 | 0.42 |
| | | | XGBoost | 0.52 | 0.07 | 0.13 | 0.43 |

### 4.1.3. Generalized Model

Samples from each of the three cities were used to create one generalized model that was then evaluated in each of the three cities. This was carried out in order to examine how models perform when we incorporate samples from different geographic regions. Table 5 provides the recall, precision, F1-dperived, and F1-macro scores for each classifier and input feature set on the test dataset.

**Table 5.** Generalized model.

|  | Input Feature | Model | Precision | Recall | F1-Deprived | F1-Macro |
|---|---|---|---|---|---|---|
| Accra | All | RF | 0.77 | 0.84 | 0.80 | 0.89 |
|  |  | MLP | 0.50 | 0.58 | 0.54 | 0.74 |
|  |  | XGBoost | 0.84 | 0.24 | 0.37 | 0.59 |
|  | User-defined | RF | 0.81 | 0.86 | 0.84 | 0.91 |
|  |  | MLP | 0.67 | 0.70 | 0.69 | 0.82 |
|  |  | XGBoost | 0.78 | 0.48 | 0.59 | 0.71 |
| Lagos | All | RF | 0.42 | 0.99 | 0.59 | 0.63 |
|  |  | MLP | 0.47 | 0.65 | 0.54 | 0.67 |
|  |  | XGBoost | 0.49 | 0.88 | 0.63 | 0.72 |
|  | User-defined | RF | 0.43 | 0.96 | 0.59 | 0.64 |
|  |  | MLP | 0.47 | 0.88 | 0.61 | 0.71 |
|  |  | XGBoost | 0.42 | 0.91 | 0.58 | 0.64 |
| Nairobi | All | RF | 0.68 | 0.92 | 0.78 | 0.73 |
|  |  | MLP | 0.72 | 0.69 | 0.70 | 0.71 |
|  |  | XGBoost | 0.76 | 0.85 | 0.82 | 0.79 |
|  | User-defined | RF | 0.71 | 0.88 | 0.79 | 0.76 |
|  |  | MLP | 0.69 | 0.87 | 0.77 | 0.74 |
|  |  | XGBoost | 0.75 | 0.78 | 0.77 | 0.76 |

Surprisingly, we observed that the generalized model has a marginally higher F1-deprived score for Accra and Nairobi compared to the individual city models for both the user and all feature sets. These results suggest that there are significant commonalities of the deprived area characteristics with the input features that may have contributed to improved performance. This may indicate that there are strong similarities between cities and that the increase in training data by combining the cities together allows the model to perform better. While it improved for these two cities, the generalized model performed poorly for Lagos, with the highest F1-deprived score of 63% (all features). This can be associated with the limited training samples for Lagos city and differences in slum characteristics, which require different features to map these slums.

*4.2. Qualitative Assessment*

The qualitative assessment focused on the generalized models because they achieve the best results, and we aim for a scalable and transferable approach. The selected machine learning algorithms are capable of predicting the probability of class categorization. Probability is the measure of the level of certainty of the prediction [64]. To obtain definite classifications of deprived and non-deprived, we fine-tuned the best threshold value for use. Fine-tuning the threshold was important for classification problems that have class imbalances, as the default 0.5 split can result in poor performance. Therefore, we defined the threshold for a city using the predicted probability values. As shown in Table 6, each city has a different threshold that better captures more deprived areas. We observed that a high threshold of over 0.7 was used for Accra and Lagos, while a low threshold was used for Nairobi. This could indicate that the model was less certain on deprived class in Nairobi compared to Accra and Lagos.

In general, the model strongly detected deprived areas compared to the reference data (Figures 4–6). Most interestingly, the model was able to differentiate between deprived areas from dense markets. This is an important finding as most studies using only satellite images fail to distinguish them because they have similar physical appearances [16,32,65]. Moreover, the model was able to map deprived areas in the suburbs. However, these areas will need field investigation since they have a unique morphological appearance compared to those in the inner city, where we have the majority of our reference data. However, we observed over-prediction for all cities (especially Lagos) for both all and user-defined features. The confusion between high-density residential buildings in close proximity to deprived

areas contributed to the majority of the overprediction. Moreover, DHS data aggregated at the enumeration area might have contributed to the over-prediction. DHS data might introduce a lot of noise because of displacement. We created an interactive map (available at https://cuer.shinyapps.io/IDEAMAPS/ (accessed on 31 October 2023)) for visualization.

**Table 6.** Classification threshold.

| City | Input Feature Set | Threshold |
|------|-------------------|-----------|
| Accra | All | 0.67 |
| | User-defined | 0.75 |
| Lagos | All | 0.69 |
| | User-defined | 0.85 |
| Nairobi | All | 0.57 |
| | User-defined | 0.55 |

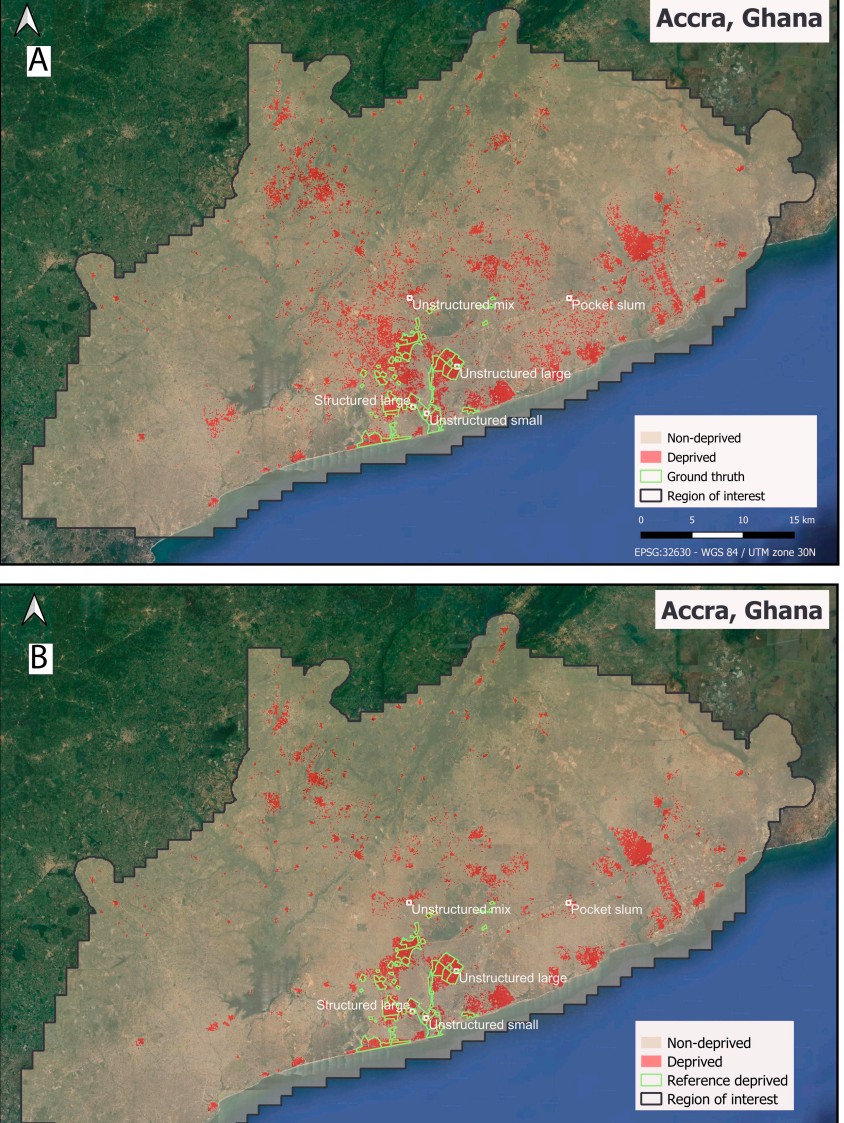

**Figure 4.** Classified map of Accra overlaid on a Google Base map. (**A**): All features (**B**): user-defined features. "Reference—deprived" is data from the Accra Metropolitan Assembly, which highlights slum locations, and these were mainly found in the inner city.

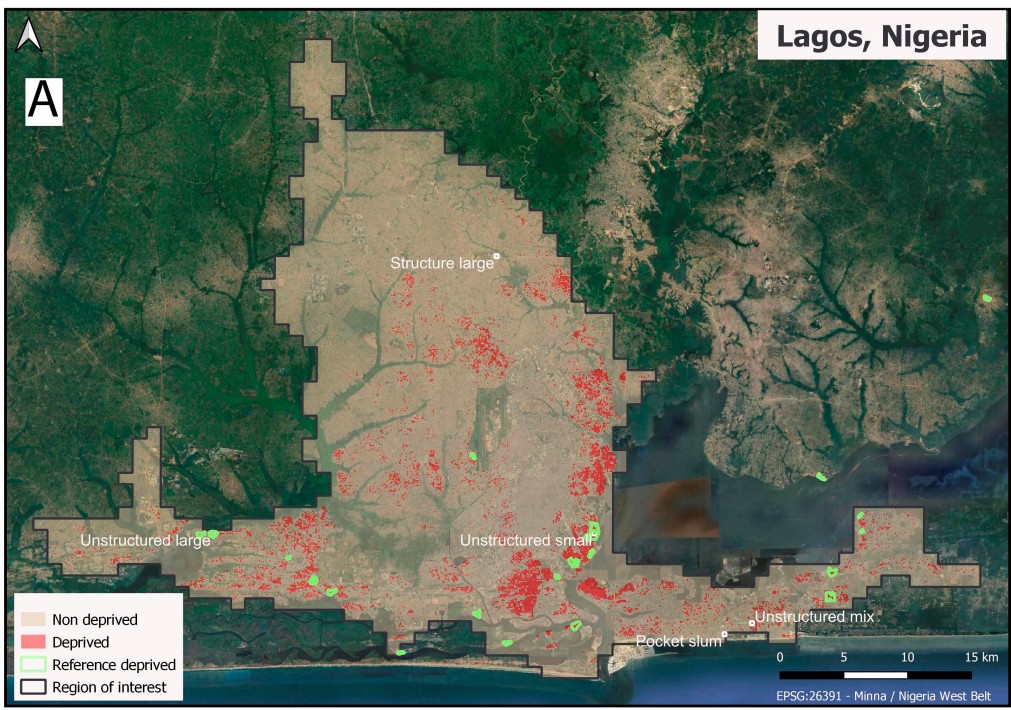

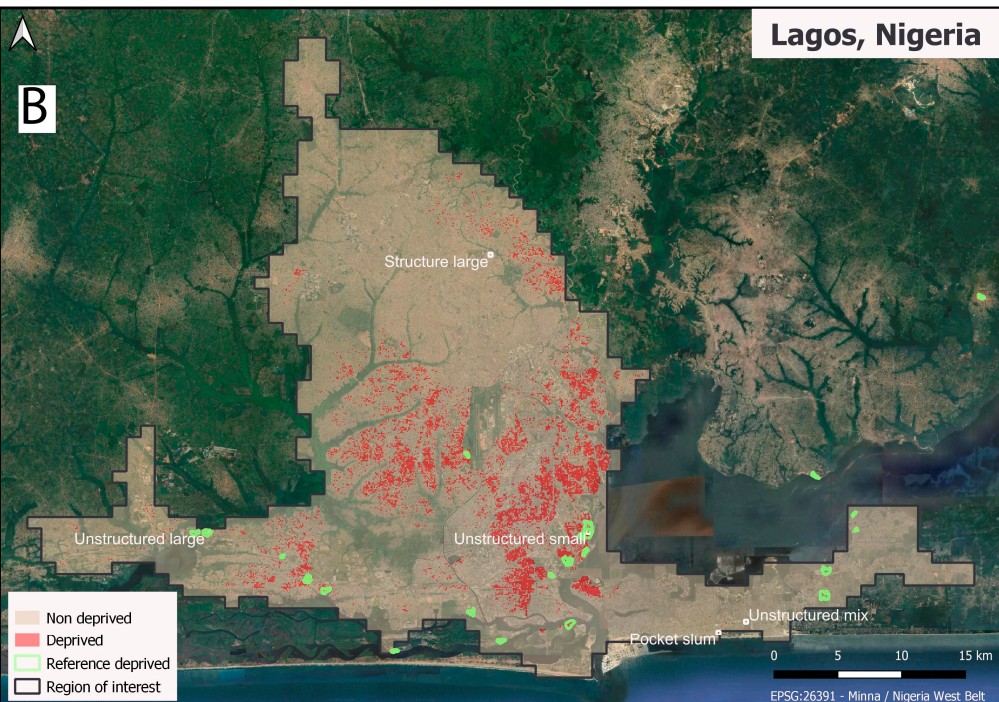

**Figure 5.** Classified map of Lagos overlaid on Google Base map. (**A**): All features (**B**): user-defined features. "Reference—deprived" is data from Slum Dweller International, which highlights slum locations.

We also assessed the model's performance to detect the deprived types (see Figure 7). Except for Lagos, large structured, deprived types were detected in Accra and Nairobi. Unexpectedly, models detected pocket slums in all the three cities. The Lagos model missed unstructured deprived mixed types due to the confusion of temporal structure, permanent structures and vegetation.

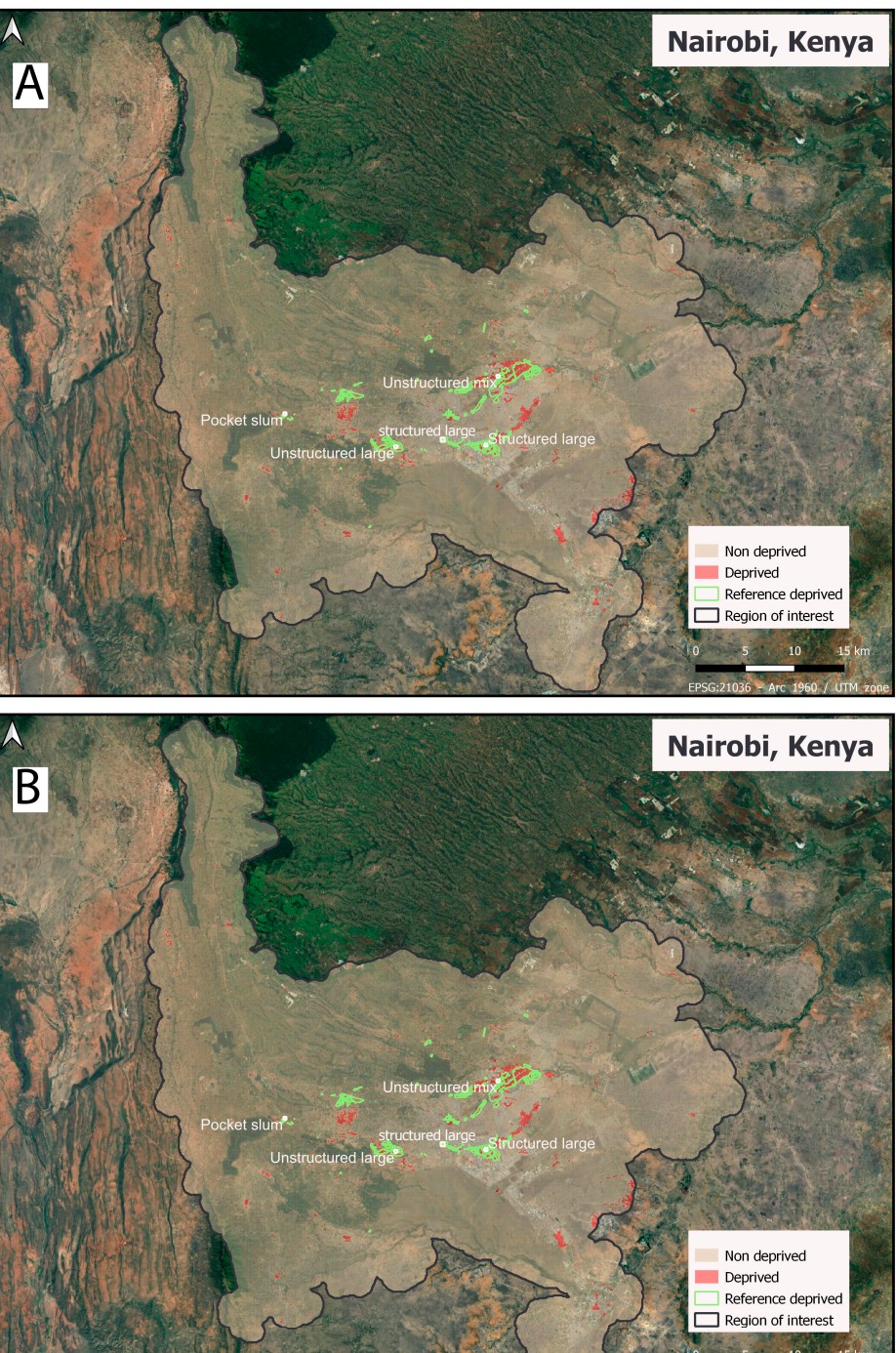

**Figure 6.** Classified map of Nairobi overlaid on Google Base map. (**A**): All features (**B**): user-defined features. "Reference—deprived" is data from Slum Dweller International, which highlights slum locations and these were mainly found in the inner city.

### 4.3. Analysis of Important Features

We calculated the Gini impurity to examine the important features and their contributions to the model, providing insights into how the model attributes different features to outputs. Figures 8 and 9 display the top 11 ranked features from the best-performing models. The feature importance score ranges from 0 to 1, with a high score indicating a strong impact on the model's predictions and a low score indicating little impact. Figures 8 and 9 show the individual city models and generalized models, respectively.

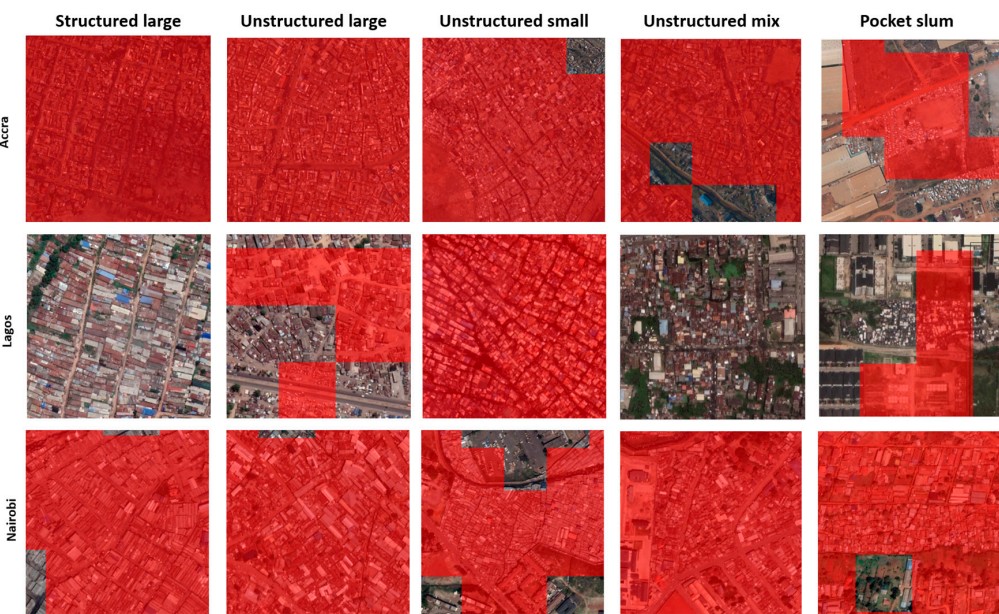

**Figure 7.** Examples of detected deprived types (in red) using the best performing model overlaid on Google Earth aerial image in Accra-Ghana, Lagos-Nigeria, and Nairobi-Kenya to assess model's performance at a citywide scale.

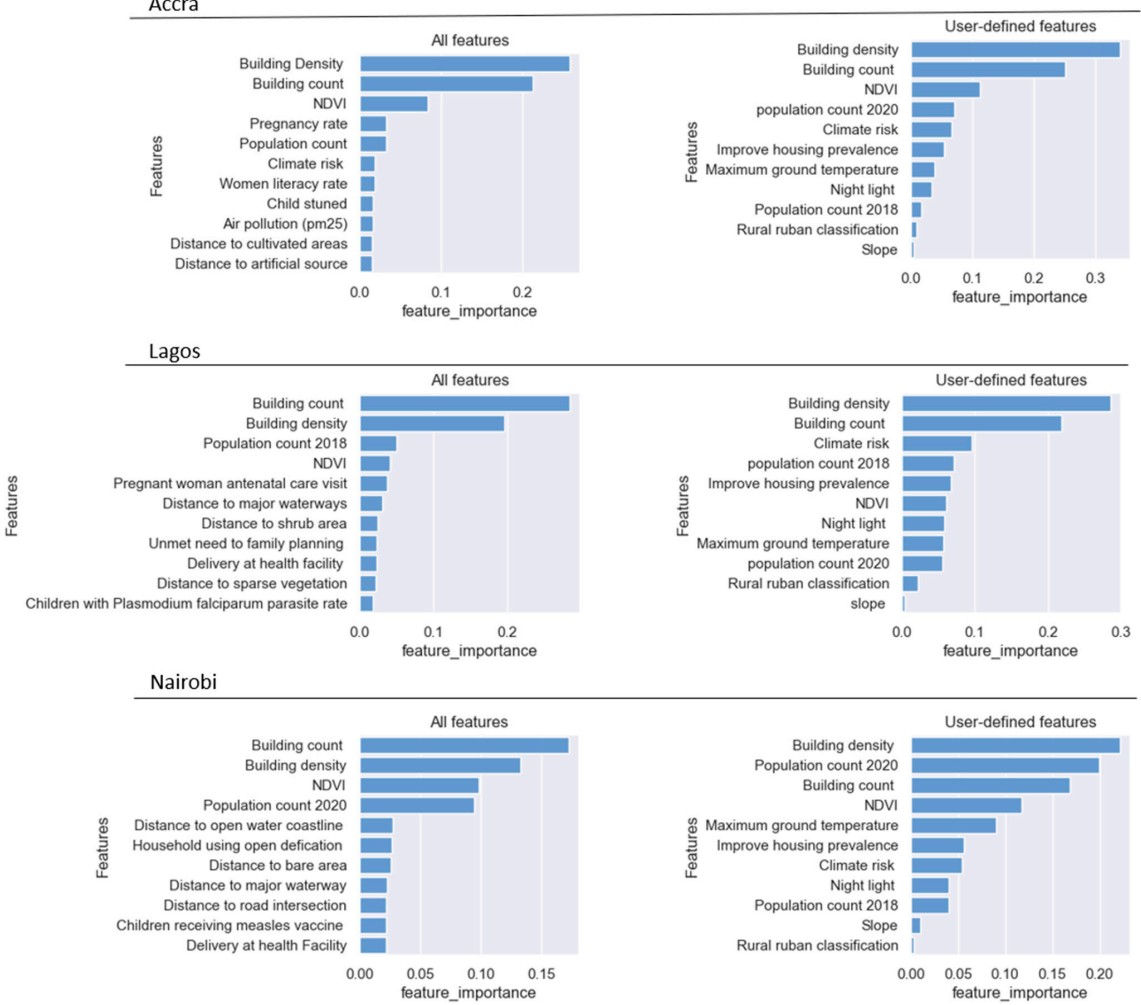

**Figure 8.** Individual city model features of importance (label descriptions in Table 2).

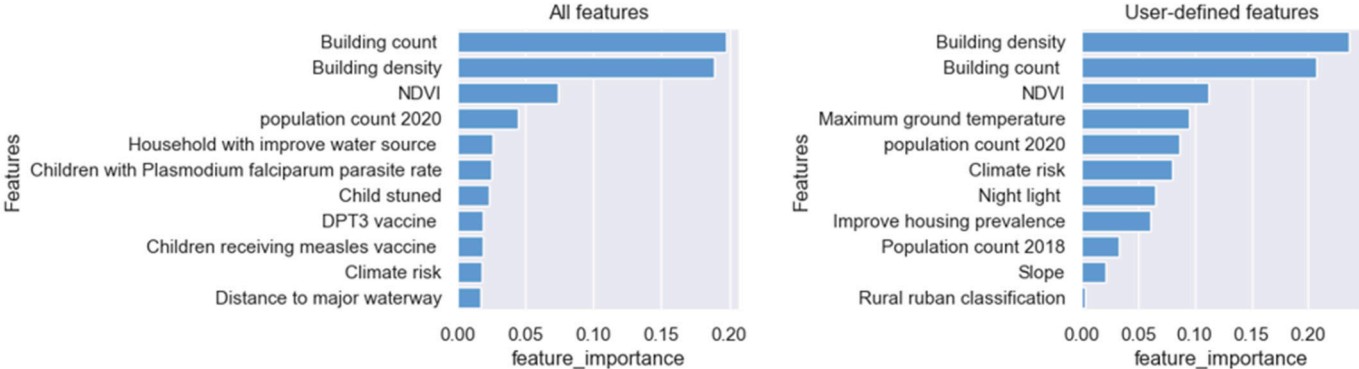

**Figure 9.** Generalized model features of importance (label description in Table 1).

Building density and count were highest in all cities, followed by NDVI and population for both all and user-defined features. These were the most common features across cities, similar to other studies [15]. Climate risk was also noted to be important in all cities. Each city tended to give different scores for every other feature. This suggests that there may be some common characteristics of deprived areas across cities, but each city also has different levels of socio-economic problems.

## 5. Discussion

This paper provides empirical evidence that open geospatial data and machine learning can provide high-resolution maps and characteristics of deprived areas. Our ML-based approach to mapping deprived areas is both scalable and transferable and could be used to generate high-resolution maps in cities using commonly available open geospatial data. Results suggest that open geospatial data allow for the modeling and understanding of many key physical and social characteristics of deprived areas across cities, providing a more holistic view of deprived areas, which has been a major criticism of existing Earth observation-based methods. To the best of our knowledge, our study is the first to extend deprived area mapping to both dense and less dense peri-urban areas using open geospatial data. These peri-urban areas are experiencing rapid urban growth and will be the future growth areas of these cities.

In contrast to other studies that rely on commercial datasets, our approach uses only openly available data and is nearly costless to scale across cities. Our model's accuracy, if not higher, is similar compared to other studies using proprietary commercial datasets. Accuracy assessment across all cities showed that open geospatial data are promising for generalizing deprived area mapping.

Notably, we have shown that our model's predictive power declined modestly (over 65% F1 score) when a model trained in one city is used to predict in another city. In spite of the differences in deprived area characteristics, model-derived features appear to identify commonalities across cities. This suggests that our approach could fill large data gaps due to poor survey coverage and could provide a crude estimate of deprived areas with little information about a city.

The generalized model using training samples from all the cities slightly improved the model's accuracy for Accra and Nairobi compared to each individual city. This indicates that these cities may have similar urban morphologies and that the models can learn from each of the cities to improve the overall results. This is encouraging as it indicates that there may be ways to stratify cities and use training data from one city to map another based on the morphological area training samples that they are derived from. Studies using satellite imagery often conclude that the large diversity of deprived areas in each city complicates modeling, thus leading to low performance [54,62]. We suspect the low variance in the input features of our model may have contributed to these results due to the general commonalities across cities.

The model performance of all features and user-defined were close for the individual city. However, the user-defined features achieved moderately high results in the city-to-city model, indicating the relevance of small input features for modeling. It also shows how too many features may be confusing, especially with limited training and validation data for optimization.

While our results are promising, this approach has important limitations. Understanding the temporal trends of deprived area characteristics is very important for researchers and policymakers to plan interventions and monitor progress. However, open data often have low temporal resolutions and are only available in specific cities or countries. Moreover, we have not yet been able to evaluate the model's ability to predict changes over time. The advance in remote sensing data (e.g., freely available Sentinel-2) can be combined with open geospatial data to map temporal changes. Satellite images can provide time series data that could possibly be used to complement socioeconomic indicators obtained from open geospatial data.

Furthermore, most of the reference data were derived from the inner cities and are limited. It was difficult to judge whether the detected deprived areas in the model in the peri-urban were actually reflective of what was actually on the ground. Future work will involve visiting and investigate these areas and gather more data for training, as there is relatively little regarding the number of features and complexity of models. The unique intra-urban diversity of deprived areas will require more training data that cover all the dynamics, both in the center city and in the suburbs.

Peri-urban areas have unique morphological and socio-cultural characteristics that require detailed investigation. We observed in the study that deprived areas in the inner city are very dense with less than 1% of areas with vegetation. Peri-urban areas tend to have a higher vegetation mix (between 10–20% vegetation). Future works will investigate these unique characteristics and develop more robust models.

This study assumes that our features were representative of the temporal resolution because urban areas do not change rapidly. The time interval of the data was 10 years. While this is reasonable, we acknowledge that the temporal differences might impact the model's performance. For example, evicted slums were not accounted for. It is worth noting that inherent errors in open geospatial data will propagate into the model, increasing the complexities of modeling.

## 6. Conclusions

The study has demonstrated that open geospatial data have the potential for mapping deprived areas. Our approach has a broad application potential across many scientific fields and may be immediately useful to inexpensively produce high-resolution maps of deprived areas to support local governments and international organizations in planning and monitoring progress towards SDG 11. Open geospatial data, by definition, are free and open, which allows other people to reproduce the results. The proposed approach combines physical and social characteristics, providing a broad view of deprived areas. Our result from the generalized model shows the ability to map at a large scale and in multiple cities. While open geospatial data has been proven to be capable of mapping deprived areas, inherent errors in the dataset potentially affect the model's performance. The study serves as an initial point to develop machine learning-based methods that combine physical and social characteristics of deprived areas beyond proof-of-concept.

**Author Contributions:** Conceptualization, M.O. and R.E.; methodology, M.O., R.E. and M.L.M. formal analysis, M.O. and R.E.; data curation, R.E. and M.O.; writing—original draft preparation, M.O.; writing—review and editing, R.E., M.K., M.L.M. and D.T.; visualization, M.O.; supervision, R.E. and M.L.M.; funding acquisition, R.E., M.K. and D.T. All authors have read and agreed to the published version of the manuscript.

**Funding:** This work was supported, in whole or in part, by the Bill & Melinda Gates Foundation INV-045252. Under the grant conditions of the Foundation, a Creative Commons Attribution 4.0 Generic License has already been assigned to the Author Accepted Manuscript version that might arise from this submission. The findings and conclusions contained within are those of the authors and do not necessarily reflect positions or policies of the Bill & Melinda Gates Foundation.

**Data Availability Statement:** The data presented in this study are available on request from the corresponding author.

**Acknowledgments:** This publication was generated as part of the IDEAMAPS Data Ecosystem project and the authors acknowledge the contributions from the stakeholders and partners communities in Lagos (Nigeria), Kano (Nigeria) and Nairobi (Kenya), as well as from the whole team including: João Porto de Albuquerque, Bunmi Alugbin, Kehinde Baruwa, Andrew Clarke, Peter Elias, Helen Elsey, Ryan Engstrom, Grace Gielink, Serkan Girgin, Diego Pajarito Grajales, Angela Abascal, Caroline Kabaria, Monika Kuffer, Oluwatoyin Odulana, Francis Onyambu, Oluwatimilehin Adenike Shonowo, Dana Thomson, Grant Tregonning, Qunshan Zhao.

**Conflicts of Interest:** The authors declare no conflict of interest.

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
