# Peer review of "Mapping Deprived Urban Areas Using Open Geospatial Data and Machine Learning in Africa"

_urbansci, doi:10.3390/urbansci7040116_

Round 1

Reviewer 1 Report

Comments and Suggestions for Authors

-Consistency in spacing after periods and commas is needed
-The use of quotation marks on Slums- define what the study means by the word earlier on and remove the marks. Also, the marks are not consistently applied
- the maps of the study area are not as informative as they should be. It would be great to indicate where the slums boundary lies as it doesn't look like the entire ROI is a slum area
-in fact not sure the featue overlaid on Google maps is the ideal way to show the results. I would consider more advnaced comparative map visuals to do this
-the data table should include a column on the original spatial resolutions as a reference to understand the need for resampling
-when used multisourced data, authors should place extra effort in expaling how the data disparities were resolved prior to analysis- particularly for ML data
-wondering if a hybrid of manual and statistically derived features would be ideal than just the manual selection
-line 415 switch physical appearance to spectral signatures 

Comments on the Quality of English Language

-several grammatical mistakes that need further proofing

Reviewer 2 Report

Comments and Suggestions for Authors

An excellent piece of research that is generally well designed, elegantly written, and well backed up by the background literature and discussion. I recommend publication with minor revisions. Revisions mainly focus on the ordering of material within the paper, with some clarification needed re. aspects of the modelling itself. Comments follow (as per line numbers in the download manuscript):

Line 88: Please give examples of accessibility features (with citations of works where these have been used).

Line 94-110: The reader may well appreciate greater clarity here re. outlining the scope of modelling that is to follow in subsection 3.6 (i.e. the individual city, city-to-city, and generalised models). This clarification should also apply to the Abstract around Line 19. The authors might consider dropping some surplus text in the earlier part of the Abstract if word count becomes an issue. Further, the modelling is barely mentioned in the Conclusion to the paper - I suggest that the Conclusion is modified to place more emphasis on this important aspect of your analysis.

Line 114: I would avoid use of the term 'ground truth data' here (and throughout). The ‘local networks’ are presumably providing good 'ground survey data' but these data will have inherent flaws too and so cannot be considered to be truth.

Line 118-9: Please do summarise here (in a few sentences) how you defined the region of interest. Also, amend citation formatting so that Thomson et al. (2021) is cited correctly (i.e. as a number, or otherwise, as appropriate to journal format).

Line 121-2: '34% of residents living in slums' statistic lacks a citation. Presumably this is number 34 from the previous sentence. Please amend.

Line 128-9: '60% of residents living in slums' statistic lacks a citation. Please amend.

Line 145, also line 190, 237, 241, 256, 259, etc: Here and on numerous occasions throughout the document there are mention of 'experts' that were consulted in this field or that. On each occasion (as appropriate) can you please specify the research group or organisation to which these experts belong (with citation(s) of appropriate works/reports so as to indicate particular expertise). I suggest defining each expert group on first mention in the manuscript to make it more efficient to refer back to each group on subsequent mention.

Line 189: The 'Data' subsection 3.2 should account for all datasets used in the research, and justify why the particular datasets were chosen. Currently this account is spread over subsequent subsections up to and including subsection 3.5. Please amend to make it easier for the reader to understand (see subsequent comments).

Line 190-1: 'recent....during the time of the research'. Please assert the temporal resolution and temporal variation of all datasets used in the research (presumably 'most recent open data available at * resolution over time span of year x to year y'). This is best summarised in section 3.2. I note the additional useful discussion of temporal resolution at the end of the Discussion section.

Line 204-6: What is 'middling' relative to? I would suggest rephrasing these sentences along the lines that the harmonised spatial resolution is useful for the purpose of your analysis (so reassert what this purpose is), and then go on to elaborate that the selected spatial resolution balances the accuracy and integration as well as the storage and computational overheads that you articulate in your text.

Line 219 & 227: Please assert that you standardised and resampled datasets to *a grid* at 100m spatial resolution, and that the vector data were rasterised and resampled to *the same grid* at 100m spatial resolution.

Line 222: Please summarise this workflow in a few sentences to give the reader some idea as to the nature of it. Amend citation formatting so that Lloyd et al. (2017, 2019) are each cited correctly (i.e. each as a number, or otherwise, as appropriate to journal format). Also note that the 2017 paper is missing in the reference list - please amend.

Line 243-8: As written these lines feel more appropriate in the discussion section (around line 495 perhaps). Perhaps rephrase or move?

Line 252: The word 'data' is always plural.

Line 252-56: These data sources should first be mentioned and discussed in 'Data' subsection 3.2 (see comment for Line 189). For clarity, please also insert a new table after table 1 to describe these data. Do assert the spatial and temporal resolution/variation and mention any related caveats for these data sources - especially the Google images.

Line 382: Please clarify in the Method section how training samples for each city model were selected for use in the generalised model.

Line 403-07: You state that the predicted probability value has been used to define the classification cut-off threshold for each city model. What are the implications of the use of these cut-off thresholds when each city model is applied to another city? And what are the implications of use of a different threshold in the generalised case? Please elaborate and be more explicit for the benefit of the reader. I suggest that your explanation here begins in each of sections 4.1.1, 4.1.2, and 4.1.3, before concluding in section 4.2 (or perhaps the Discussion).

Comments on the Quality of English Language

The quality of English language and grammar is high, with occasional unusual misspelling (e.g. 'temporal' instead of 'temporary', 'deprived' instead of 'deprivation' types - both in section 3.7) / wrong tense used. Suggest that the authors proof read again.

Round 2

Reviewer 2 Report

Comments and Suggestions for Authors

I have reviewed the manuscript and author response to the first review. I believe that the manuscript has been sufficiently improved to warrant publication in Urban Science.